# Microplastics: A Matter of the Heart (and Vascular System)

**DOI:** 10.3390/biomedicines11020264

**Published:** 2023-01-18

**Authors:** Elisa Persiani, Antonella Cecchettini, Elisa Ceccherini, Ilaria Gisone, Maria Aurora Morales, Federico Vozzi

**Affiliations:** 1CNR Institute of Clinical Physiology, via Giuseppe Moruzzi 1, 56124 Pisa, Italy; 2Department of Clinical and Experimental Medicine, University of Pisa, via Volta 4, 56124 Pisa, Italy

**Keywords:** microplastics, nanoplastics, environmental exposure, cardiovascular system, heart, contaminants

## Abstract

Plastic use dramatically increased over the past few years. Besides obvious benefits, the consequent plastic waste and mismanagement in disposal have caused ecological problems. Plastic abandoned in the environment is prone to segregation, leading to the generation of microplastics (MPs) and nanoplastics (NPs), which can reach aquatic and terrestrial organisms. MPs/NPs in water can access fish’s bodies through the gills, triggering an inflammatory response in loco. Furthermore, from the gills, plastic fragments can be transported within the circulatory system altering blood biochemical parameters and hormone levels and leading to compromised immunocompetence and angiogenesis. In addition, it was also possible to observe an unbalanced ROS production, damage in vascular structure, and enhanced thrombosis. MPs/NPs led to cardiotoxicity, pericardial oedema, and impaired heart rate in fish cardiac tissue. MPs/NPs effects on aquatic organisms pose serious health hazards and ecological consequences because they constitute the food chain for humans. Once present in the mammalian body, plastic particles can interact with circulating cells, eliciting an inflammatory response, with genotoxicity and cytotoxicity of immune cells, enhanced haemolysis, and endothelium adhesion. The interaction of MPs/NPs with plasma proteins allows their transport to distant organs, including the heart. As a consequence of plastic fragment internalisation into cardiomyocytes, oxidative stress was increased, and metabolic parameters were altered. In this scenario, myocardial damage, fibrosis and impaired electrophysiological values were observed. In summary, MPs/NPs are an environmental stressor for cardiac function in living organisms, and a risk assessment of their influence on the cardiovascular system certainly merits further analysis.

## 1. Introduction

Plastic demands raised enormously during the last few years. Plastic products bring several benefits to human beings and the community, playing an even more fundamental part in the food industry, constituting a sustainable and safe option. This widespread plastic use has inevitably produced plastic waste. In 2020, nearly 30 million tonnes of plastic waste were collected in European countries, of which only 34.6% undertook the recycling route (https://plasticseurope.org/, 17 January 2023). Unfortunately, a still-too-high percentage of post-consumer plastic waste is sent to landfill or misprocessed, contributing to ecosystem pollution. The persistence of current production and waste management practices will predictably lead to 12 billion tonnes of plastic waste in the natural environment by 2050 [1]. In addition, only in 2010, up to 12.7 million metric tonnes of plastic were released into the oceans [1], leading to the accumulation of over 250,000 tonnes of plastics abandoned in oceans [2]. In this scenario, it has been foreshadowed that plastic waste in the environment will constitute a stratigraphic marker for the Anthropocene [3]. Plastic waste persists in the natural environment not only for its vast problematic presence itself but also for its persistence and slow degradation. These environmental pollutants pose severe threats to different animal species, constituting physical traps, barriers to the food supply or congestion and ultimately causing death [4,5].

Plastic pollutants which enter the ecosystem in small pieces, typically less than 5 mm in size, are called microplastics (MPs) and are generally industrially produced. In such cases, where plastic particles of this precise size are intentionally made, pollutants are called primary MPs [6]. The small particles generated by degradation processes such as thermo-oxidation or mechanical segregation on larger plastic pieces are called secondary MPs [7]. Additionally, plastic waste can present as particles of less than 1 μm, called nanoplastics (NPs) [8,9]. MPs and NPs are chemically inert compounds which are able to penetrate and indefinitely persist in both terrestrial and marine environments and freshwaters [10,11,12], concurring to a recently described ecotoxocity [13,14]. They were recently reported as health-hazardous compounds [15] due to their ability to act as vectors for chemical contaminants such as heavy metals [16,17] as well as pathogens [18,19]. 

Due to their small size, they directly enter the food chain by the gastrointestinal route, as demonstrated by the ingestion of contaminated algae by planktonic crustacean *Daphnia magna*, then to a secondary consumer fish, and finally reaching an end consumer fish [20,21,22]. 

Additionally, it has been speculated that MPs and NPs might act as carriers for pollutants to living organisms, thus leading to their bioaccumulation [23,24]. Regardless of the model under study, it is clear that MPs and NPs can be ingested and accumulated in larger marine fauna by direct transfer from prey to predator, including big predators like humans. Ingestion is the main route via which plastics enter the human body [25].

Sources of human-ingested MPs and NPs are mussels [26], commonly consumed fish [27], commercial salt [28], sugar and honey [29], tea bags [30] and drinking water [31,32]. Unsurprisingly, contaminants can also be found in plastic water containers and bottled water [33,34] and, massively, on plates during meals after settling of dust particles [35]. In vivo murine models showed that MPs, upon intestinal internalisation, reach the bloodstream within 15 min after ingestion, mainly accumulating in the liver [36,37]. Exposure of hepatocytes to MP compromises cell membrane integrity in a dose-dependent manner, as demonstrated in the HepG2 cell model [38]. Using a gastric adenocarcinoma cell line, it was also possible to observe changes in cell shape and the triggering of an inflammatory response which may ultimately lead to cell death [39]. Furthermore, recently it has been described that NPs can enter the human and livestock food chain through plant consumption [40]. Toxicity might be directly caused by plastic debris by their accession to plant cells. In fact, it has been demonstrated that plant cells can take up particles of less than 0.1 μm, thus entering the food chain and representing a serious threat to higher organisms. Moreover, due to the compounds and pollutants adsorbed on the surface of MPs/NPs, they could inhibit plant growth and imbalance plant symbiosis with fungi and microbial species [40].

Another known route of exposure to MPs is inhalation, as they are consistently found in the air. It was recently shown cytotoxicity, ROS production, and inflammation in an in vitro lung cell model [41]. In a study including healthy and asthmatic mice treated with MPs via inhalation, the authors demonstrated severe lung inflammation and worsening respiratory failure in the latter [42]. Additionally, dose-dependent pulmonary fibrosis has been recently observed in murine models, activating oxidative stress markers upon polystyrene administration [43]. 

Regardless of the different routes via which MPs and NPs may access the human body, they undertake internalisation through active or passive influx, and they reach distant organs using the circulatory system. This review provides a picture of the latest pieces of evidence of the effects on the vascular system and cardiac tissue exerted by MPs and NPs in aquatic fauna and humans (schematically summarised in Figure 1 and Figure 2). The cardiovascular system, also if not the main target in terms of accumulation, is the main route of distribution for MPs and NPs. Recent works presented in this review show the first detrimental effects of plastic pollution on cardiovascular function. 

## 2. MP/NP Effects on Aquatic Fauna Vascular System

Whenever plastic materials reach the aquatic environment, they become prone to disintegration through the mechanical action of waves and/or water flows, water animals, and UV radiation. Water is, therefore, the vector for pollutants, namely plastics, and the way of access to the aquatic vertebrates and invertebrates’ cardiovascular systems.

Fishes have a closed cardiocirculatory system, where blood circulates within vessels of various sizes and does not fill any cavities. It is characterised by the presence of one heart, is very simple in structure, and is generally localised behind the gills. Oxygenated and nutrient-rich blood is pumped by the heart in all parts of a fish body within arteries; from peripheric tissues, deoxygenated and nutrient-poor blood is transported through veins towards the different components of the heart and reaches the gills again. In this context, gas exchanges are facilitated, incorporating oxygen from water and eliminating carbon dioxide from the body. 

In 2016, Lu and collaborators found polystyrene (PS) microparticles sized 5 μm and 20 μm (exposure concentration 20 mg/L)deposited in gill filaments of *Danio rerio*, which can then possibly be transferred to the capillaries [44]. Consistent with these findings, Ding et al. not only confirmed the presence of PS-MPs in the gills of red tilapia (*Oreochromis niloticus*) at all dispersed concentrations of 1, 10 and 100 mg/L, but this organ showed the highest bio-accumulation [45]. PS-MPs deposition in gills was 18.4, 49.4 and 71.7 × 10^4^ μg/kg for nominal oral exposures of 1, 10 and 100 μg/L. Furthermore, the same study observed the distribution of these particles in the circulatory system and the consequent presence in other peripheral organs, such as the gut (concentrations of 25.5, 89.9 and 171.1 × 10^4^ μg/kg), liver (12.8, 28.7, 36.6 × 10^4^ μg/kg) and brain (10.6, 32.5 and 40.5 × 10^4^ μg/kg) [45]. 

Previous evidence in a type of Mytilus reported that ingested 0.51 g/L PS-MPs could pass from the gastrointestinal cavity to the haemolymph and subsequently be distributed to other tissues through the circulatory system [46]. Molluscs present with an open circulatory system where blood is pumped by the heart into the body cavities, and tissues are surrounded by blood so MPs can be transported directly to all major organs. In mussels, it seems that the accumulation mechanism is similar, regardless of the size of plastic particles, possibly via translocation. In two experiments, it was possible to detect them directly in the haemolymph and inside haemocytes after a 3-days exposure time [46,47]. Particle size appears to influence the capacity of PS to translocate from the intestinal tract to the haemolymph, with the majority of smaller particles found in the circulation [46]. These studies showed that the persistence of MPs in fishes’ circulatory systems has some major implications for predators and end-consumers species, including humans.

In common carp (*Cyprinus carpio*), exposure to MPs alters blood biochemical parameters, i.e., reduced plasma levels of acetylcholinesterase (AChE) and gamma glutamyl-transferase (GGT) and increased aspartate aminotransferase (AST), alanine aminotransferase (ALT), alkaline phosphatase (ALP) and lactate dehydrogenase (LDH) [48]. Additionally, in immunological tests, specifically lysozyme and alternative complement (ACH50) activities, total immunoglobulins and complement C3 and C4 factors are lowered [48]. MPs (and NPs) are known to interact with heavy metals, and 30-day exposure to a mixture of plastic debris, mainly polyethylene (PE) and cadmium, further enhanced the detrimental effects on the biochemical and immunological assets in carp [48]. Importantly, these effects were observed for both concentrations tested (250 and 500 μg/L) when combined with the heavy metal, with the only exception of LDH, which significantly changed at the highest quantity tested. These results suggest a perturbation in cellular homeostasis due to the changes in biochemical parameters, and the altered immune system components might predispose to infectious diseases. To note, this work used a mixture of plastic fragments, mainly represented by PE, the most common type of characterised plastic in water, followed by polypropylene (PP) and PS [7].

In addition, PS orally administered accumulated in the gill with ROS production and histopathologic changes in loco in both 20 and 200 mg/L treated groups. Concentrations of MPs found in gills were 39.06, 73 and 175 item/individual for 2, 20 and 200 mg/L treated groups. This bioaccumulation was similar to what was found in the intestine. Structural damages were also observed in other distant organs demonstrating the circulatory system’s crucial role in transporting MP [49]. 

Recently, Sun et al. investigated the toxic cardiovascular effect of NPs in zebrafish embryos. The fluorescence images demonstrated that the NPs could inhibit sub-intestinal angiogenesis, which can be directly related to impaired cardiovascular formation and development and ultimately suggest that NPs can damage the vascular endothelium in these organisms [50]. Thrombotic effects were also evaluated, and a hypercoagulable state was found in the caudal vein, where authors observed erythrocytes aggregation and neutrophils recruitment and an incidence of thrombosis markedly elevated after NPs exposure [50]. From a hemodynamic perspective, decreased carbon monoxide levels and blood flow rate in treated groups were also dose-dependent, clearly detectable at concentration exposures of 100 and 200 μg/mL [50]. Vascular resistance increases after thrombosis, leading to elevation in blood pressure directly and the contraction process of the myocardium, thus affecting carbon monoxide levels. Therefore, hemodynamic changes further aggravate the thrombotic process. The same study also found ROS production and a generalised inflammatory response for 100 and 200 μg/mL concentrations. Researchers concluded that NPs trigger a pro-inflammatory and pro-coagulant state through the vascular endothelial cell layer, ultimately promoting thrombosis in vivo [50]. Analysis of haemocytes of the marine bivalve *Mytilus galloprovincialis* revealed blood toxicity of increasing concentrations of PS-NPs (1, 5 and 50 μg/mL) [51]. In the treated group, induction in cellular apoptosis was detected, with a rapid lysosomal destabilisation evidenced by an enhanced lysosomal enzyme release. The latter effect was more pronounced at the highest concentration of 50 μg/mL, accompanied by decreased phagocytic activity, indicating that the immunocompetence is compromised with PS-NPs [51]. Interestingly, 5 and 50 mg/mL administration induced an immediate lysozyme release, whereas a comparable increase was observed at all the concentrations tested for longer incubation times [51]. These results undoubtedly demonstrate that NPs compromise the immune function in marine invertebrates.

## 3. MP/NP Impact on Heart Physiology in Aquatic Organisms

An assessment of MP damage of vital organs, the heart in particular, might be deducted in simpler organisms such as molluscs, given the open circulatory system where the blood cells can circulate directly to all major organs. This system allows the direct study of plastic particle uptake, translocation, bioaccumulation, and, possibly, the mechanisms of toxicity. Hence, studies investigating the impact on the fish heart of other chemical pollutants widely present in freshwaters and seas, such as MPs and NPs, are essential not only for the well-known ecological issues but also for the implications on aquatic biota and the end-consumers such as humans.

The disturbance of cardiac physiology during morphogenesis caused by toxic chemical contaminants such as dioxins, polychlorinated biphenyls (PCBs), and polycyclic aromatic hydrocarbons (PAHs) has been well-established and reviewed in [52]. Cardiotoxicity is detectable as oedema and changes in permanent ultrastructure, affecting swimming and feeding performances later in life. Thus, considering the strong influence of oedema in developing fish hearts, it has been used as an indicator of cardiotoxicity in fish larvae and embryos.

Corroborating this hypothesis, pericardial oedema was visible in zebrafish embryos from NP-exposed parents. In addition, embryos presented with yolk sac malformation, suggesting a generally toxic effect on the cardiovascular system when exposed to concentrations of 100 and 200 μg/mL [50]. In zebrafish larvae and embryos from maternal and/or co-parental exposure groups, PS-NPs also affected heart rate when ingested at 120 mg/mL [53]. The authors speculated that the observed bradycardia might result from the PS-NPs interaction with the cardiac sarcomeres. The pericardium was unaffected in this study, probably because of the dose-dependent NP deposition in this membrane [53]. Consistent with this, another study on zebrafish larvae reported the PS-NP localisation in the pericardium only at the highest concentration of 10 ppm [54]. In this study, upon waterborne exposure, PS-NPs accumulated in the yolk sac within 24 h post-eggs-fertilisation and migrated not only to the developing heart but also to the gastrointestinal tract, gallbladder, liver, pancreas and brain. Notably, exposed groups showed significant bradycardia when compared with controls, even at the lowest concentrations of 0.1 ppm of PS-NPs [54]. Another recent work confirmed that parental exposure to 20 μg/L PS-MPs considerably decreased heart rate in marine medaka offspring [49]. In addition to speculated interactions of particles with cardiac sarcomeres, the influence on heart rate might be explained by the oxidative state generated by MPs.

Tachycardia, instead, was noted in goldfish larvae exposed to PS-MPs and -NPs mixture even at low concentrations (10 μg/L; 100 and 1000 μg/L were also tested) after 7 days of treatment [55]. These apparently opposite effects on abnormal heart rate are possibly due to the susceptibility of this organ to the oxidative stress generated by the introduction of plastic toxicants in the body.

Figure 3 and Table 1 and Table 2 offer an overview of the main features of MPs and NPs studies on the cardiovascular system of aquatic organisms included in the present review.

## 4. MP/NP Effects on Mammalian Circulating Cells and Vascular System

Following plastic particles’ accession to the body via different routes, i.e., ingestion, inhalation, or dermal, MPs and NPs are transported through the circulation. Therefore, understanding how blood cells interact with these pollutants might have significant implications for humans. In particular, the interaction of MPs/NPs with serum proteins might reveal information regarding the bioavailability and distribution of these compounds.

In a study, PS-NPs—with different surface functionalisation were synthesised (80–170 nm diameters, number of particles per mg of serum protein ranging from 4 × 10^10^ to 2 × 10^12^), incubated with human serum, and uptake was analysed. These particles were taken up by monocytes, granulocytes and myeloid dendritic cells but not by lymphocytes [56]. The uptake of PS-NPs selectively by antigen-presenting cells indicates that the mechanism of internalisation to the cytoplasm may occur through phagocytosis rather than non-specific cellular uptake. In a study on whole blood samples, lymphocytes exhibited very low internalisation levels, regardless of time and dose of exposure to PS-NPs, whilst there was a time-dependent effect on the uptake at lower concentrations (1, 10, 25 μg/mL) for all white blood cells and monocytes [57]. Another study using whole blood proved a high internalisation of these particles in monocytes and granulocytes, whereas all other particles are only taken up in significant amounts in monocytes. Additionally, authors speculated that the biodistribution of these particles might also depend on size, charge, and reactive groups on the surface [56]. Upon vein tail injection, functionalised PS particles were found in lymphatic and non-lymphatic organs, showing high stability for these plastic species in human serum [56].

A study focused on PS-MPs and NPs internalisation in phagocytes showed that monocytes, grown in suspension, ingested almost 100 times less particles than macrophages. This is probably due to the greater membrane surface area of suspension cells when exposed to a concentration of 20 μg/mL of reactive PS-particles [58]. Larger plastic particles boosted phagocytosis and induced the secretion of IL-6, implying that the release of inflammatory cytokines induced by MPs may be linked to phagocytosis [58].

Under physiological conditions, particles are surrounded by plasma proteins, forming a corona [59]. Coated particles were internalised to a lower degree, as the coating may prevent the interaction of NPs with plasma membranes by reduction of surface reactivity [58]. MPs-coronas, instead, displayed an increased cellular uptake by phagocytes [58], possibly involving other energy-independent uptakes different from phagocytosis and endocytosis. Corona formation on NPs and protein-induced coalescence of NPs are known to cause conformational changes and protein denaturation, leading to the formation of a non-biocompatible complex, promoting cytotoxicity and genotoxicity on blood cells to a greater extent when compared to the virgin-NPs [60]. Numerous plasma proteins displayed strong affinity towards NPs. They generated a corona expected to escape from the immune system, prolong persistence in circulation and interfere with cellular and molecular processes [60].

Upon internalisation, PS-NPs have been shown to exert a strong genotoxic effect on some blood cells, evident as DNA breakage, specifically in monocytes and in polymorphonuclear cells (PMNs), even if the internalisation rate did not correlate with genotoxicity [57]. Massive DNA damage was also observed in lymphocytes exposed to coronated-NPs when compared to virgin-NPs. This effect, evident at concentrations of 5 and 7.5 μg/mL, was associated with the cytotoxic and haemolytic effect [60]. This could imply that genotoxicity is influenced by plasma proteins, turning into bio-incompatible and leading to harmful biological activity.

Red blood cells (RBCs) are blood’s dominant cells, and the biological impact of plastic particles on these entities may be deducted from observations using nanoparticles. RBCs treated with PS-NPs triggered the formation of aggregates and stimulated the adhesion to endothelial cells [61]. These effects were more pronounced with increasing concentrations of PS-NPs, with evident differences in the range of 0.05–0.5 mg/mL tested and decreasing their size [61]. Since RBC adhesion to vascular endothelial cells is considered a potent effector in cardiovascular diseases, NPs might be considered a potential risk factor. Another study focused on potential adverse outcomes in RBC treated with silica nanoparticles (sNPs). Larger sNPs adsorbed to the RBC surface induce a local membrane deformation, while smaller sNPs do not show similar effects [62]. The larger the sNPs, the higher the particles’ haemolytic activity and internalisation rate observed at both 50 and 100 μg/mL mounts tested [62]. These results imply that plastic particle sizes influence cardiovascular cellular components differently.

PP particles could increase the haemolytic effect, more pronounced using larger particles rather than smaller ones, in a concentration-dependent manner [63]. In fact, haemolysis was observed in larger particles (25–200 μm) in both DMSO-vehicle and dispersed powder particles, at concentration ranges of 10–100 or 300–4500 μg/mL, respectively. As regards pro-inflammatory cytokines release, IL-6 secretion changed significantly with PP particles at a concentration of 100 μg/mL at the smaller size, suggesting that smaller PP particles could mimic potential pathogens, whilst IL-2 secretion was observed after treatment with larger PP particles [63]. In addition, the secretion of TNF-α increased with smaller PP particles, particularly at higher concentrations (100–1000 μg/mL), implying that smaller PP particles have the potential to trigger inflammation [63]. Conversely, temporary immunosuppression has been described in vivo after early-life exposure to combustion-derived particles (200 μg/m^3^) in mice because of suppression of dendritic cells, production of IL-10 (anti-inflammatory cytokine), and inhibition of T-helper type 2 cells [64].

The common feature when MPs are in circulation seems to be an inflammation state, vascular occlusions, and hypercoagulability, at least in a rat model [65]. PS particles (0.15 mL/100 g body weight) were used as a trigger to experimentally induce pulmonary embolism [65].

Human umbilical vein endothelial cells (HUVEC) represent an excellent model for studying vascular endothelium properties. Recently, a study reported the effects of PS-MPs on HUVEC to shed light on their impact on endothelial cells’ functionality and vasculature formation [66]. MP exposure was a significant risk factor for endothelial dysfunction and vasculature malformation. Over a short period, contact with PS-MPs at a concentration of 80 μg/mL impaired angiogenesis through mitigation of the vascular endothelial growth factor (VEGF) signalling pathway, whilst a more prolonged exposure affected cell viability through the activation of autophagy and necrosis [66].

Overall, these results consolidate the hypothesis that MPs/NPs can induce cardiovascular toxicity and they directly cause heart dysfunction. Even if the mechanistic effect has not been fully clarified, the data present in the literature provide a basis for a new factor favouring cardiovascular disease onset.

## 5. MPs/NPs Effects on Cardiac Physiology in Mammals

Studies on mammals demonstrated that nanoparticle exposure to the cardiovascular system negatively impacted cardiac function [67,68]. Air pollutants, such as particulate matter, promote cardiotoxicity, compromising vascular function, increasing blood pressure, and promoting myocardial infarction, thus highlighting that they represent a risk factor for cardiovascular disease. Therefore, investigating the potential for MP/NPs role in cardiovascular toxicity may have enormous social and economic benefits.

A number of studies exploring bioavailability and biodistribution in mammals detected the deposition of plastic fragments in cardiac tissues. Studies found in heart PS-MPs (30 mg/kg body weight) [69] or PS-MPs differently charged (125 mg/kg body weight) after oral administration [70]. In particular, PS-MPs administered through the oral route were found mainly in the blood (135.86 μg/mL tissue), spleen (106.31 μg/g tissue), and lung (103.70 μg/g tissue), and also in the heart (45.35 μg/g tissue), [69]. Acute [71] and chronic [72] oral exposure reported neither PS-MPs in the heart nor morphological changes in mice. As regards the acute exposure, researchers intragastrical-inoculated mice with 50 mg/mL of 60 nm-diameter plastic fragments [71], whilst for the repeated exposure, authors reported no effects for all concentrations and sizes of particles tested (1.25, 25 and 34 mg/kg body weight for 1, 4 and 10 μm-size respectively) [72]. Interestingly, upon inhalation, authors identified PS-NPs in the cardiac tissue of pregnant rats and foetuses after acute exposure [73]. Particularly, significant nanopolystyrene deposition was detected not only in the maternal heart but also in the lung and spleen, whilst the deposition of PS-NPs was also elevated in all foetal tissues analysed [73]. Inhalation of multi-walled carbon nanotubes (MWCNT) commonly used for textiles as well as for body and vehicle protection causes coronary arteriolar endothelium damage in young adult male rats [74].

Information regarding MPs impact on heart ultrastructure derives from a study elucidating PS-MPs cytotoxicity in rats following ingestion [75]. Transmission electron microscopy was applied to directly visualise PS-MPs, which were found to be internalised in cardiomyocytes, confirming that they were translocated to the heart via the circulatory system. Structurally, the heart presented with damage and extensive apoptosis of the myocardium at the experimental doses of 5 and 50 mg/L of PS-MPs, with collagen synthesis. On a molecular basis, PS-MPs exposure at high concentrations (50 mg/L) considerably increased myocardial creatine-kinase MB and cardiac Troponin I, two important markers of myocardial damage. Additionally, oxidative stress evaluation revealed a decrease in an antioxidant molecule in favour of pro-oxidation in the 5 and 50 mg/L-treated groups, and the activation, for the highest dose, of the Wnt/β-catenin signalling pathway, a promoter of fibrosis [75]. These results suggest that PS-MPs trigger cardiotoxicity through cardiomyocyte apoptosis driven by oxidative stress and shifting to cardiac fibrosis mediated by Wnt/β-catenin.

PS-MPs in circulation were intravenously introduced at 0.15 mL/100 g body weight and directly caused pulmonary embolism in a rat model with permanent pulmonary obstruction [65]. Compared with healthy controls, rats subjected to PS-MPs demonstrated significant arterial hypotension and increased arterial lactate concentration, which may indicate hypoxia [65].

The acute effects of PS-NPs arose from a study on rat neonatal cardiomyocytes [76]. Positively charged PS-NPs at a concentration of 25 μg/mL were rapidly internalised into neonatal rat ventricular myocytes in the presence of electrical pulses to synchronise the cardiac contraction in vitro. A significant decrease in intracellular calcium levels was observed, inhibiting the L-type calcium channel (LTCC). This effect, together with a reduction in electrophysiological performance in neonatal cardiomyocytes, caused a decrease in contraction forces in the early phase. During the late phase, mitochondrial membrane potentials were remarkably lowered with impaired glycolytic homeostasis and lower basal oxygen consumption rate, affecting ATP production [76].

Taken together, these results emphasise the considerable risks connected with the escalating MPs/NPs pollution for the hearts of human infants and adults and the potential threat of cardiovascular diseases.

Figure 4, Table 3 and Table 4 offer a summary of the data on MPs/NPs effects on the mammalian cardiovascular system.

The dermal route of MPs/NPs exposure can be considered a less important way of accessing the mammalian CV system. Humans can interact with plastic debris through the skin during washing with contaminated water or by using cosmetic products. Little is known about the mechanisms involved in the internalisation of plastic particles via dermal contact, but it has been described the penetration of striatum corneum for particles below 100 nm [77]. In addition, evidence on human epithelial cells showed enhanced oxidative stress from exposure to MPs/NPs; therefore, it is reasonable to speculate that the dermal route might be susceptible to the presence of plastic fragments and be relevant for their accession in circulation.

## 6. Discussion, Conclusions, and Recommendations

Nowadays, MPs/NPs represent a novel class of environmental pollution and are ubiquitously distributed in the global biosphere. In addition to physical, biological, and chemical insults, plastic fragments challenge the immune system, representing an additional stressor. Observation in simpler organisms as well as in mammal circulating immune cells demonstrated a common impairment in cellular homeostasis and alteration in blood biochemical parameters. These effects result in health hazards and predispose affected organisms to infectious diseases, potentially influencing the survival rate. MPs/NPs interaction with antigen-presenting cells in the mammal cell system can trigger a cytotoxic and genotoxic effect. Plastic particles’ interaction with plasma proteins might prolong their persistence in circulation, thus interfering with cellular homeostasis and escaping immune system surveillance.

RBCs are the most represented and long-lived cells in the blood, and plastic particles showed internalisation and persistency within these cells. This aspect might be exploited as a carrier for targeted drug delivery of nanoparticles, but, on the other hand, it might trigger RBC adhesion to the vascular endothelium, a key feature of thrombus pathogenesis. Thrombosis is a critical health concern worldwide, identified as a pathological factor for cardiovascular disease and thrombotic complications.

The state-of-the-art of MPs/NPs demonstrate that environmental exposure via different routes to plastic particles could undoubtedly affect blood immune system cells, leading to genotoxicity and influencing heart physiology. Different plastic types were found to accumulate in the heart, describing a trophic transfer via the bloodstream, and cardiotoxicity was observed in mammalian systems and fishes. Plastic fragments could affect the cardiovascular system at different points in a fish’s life cycle. In fact, upon maternal transfer, alterations in heart rate were observed in embryos and larvae, and morphological changes were also observed. In mammals, reported studies showed impaired heart contractility, neonatal cardiomyocyte apoptosis, and activation of fibrotic processes. Taken together, these data suggest that MPs/NPs interaction with a developing heart impairs cardiac function, ranging from severe loss of function and complete failure of cardiac morphogenesis in early stages to alterations, in the form of arrhythmia or reduced contractility, in developing and likely adult hearts. These observations pose major concerns about neonatal cardiomyocytes’ susceptibility to plastic pollution, suggesting that actions need to be taken to limit the misuse of potentially hazardous plastic waste that can be inadvertently introduced into breast milk, air, water, and food chain, affecting the offspring.

A risk analysis is fundamental for clarifying the extent of cardiovascular toxicity in an adverse outcome framework in humans. In the absence of epidemiological data, in vitro/ex vivo toxicological studies might be conducted in aquatic animals per se, broadening our awareness of the biological effects of MPs/NPs on simpler organisms, such as the influence factors, namely size, shape, and composition of MPs, and physiological mechanism involved in the cardiovascular toxicity. This work should include molecular initiating events and consequent pathways involved. Understanding the mechanisms for alterations in the cardiovascular environment caused by plastic fragments in aquatic organisms may have direct implications for human well-being as MPs/NPs may be transferred further up the food web, reaching the top-level consumer, humans.

MPs/NPs are an environmental stressor for cardiac function in living organisms, further aggravated by a plastic affinity for pathogens, heavy metals, and other persistent organic pollutants. Among this group of hazardous compounds, polycyclic aromatic hydrocarbons were observed to disrupt ionic conductance in fish cardiomyocytes, with cardiac morphogenesis defects likely to occur during waterborne exposure [78].

To establish the extent to which the cardiovascular system is affected by plastic pollution in natural populations and to define the mechanistic effects, some aspects need to be improved for in vitro and/or in vivo studies:Refinement of techniques to quantify small plastic fragments.Identification of plastic type, size, shape, and charge causative of adverse effects mainly in the heart. Weathering/ageing of plastic particles may be considered.Realistic time of exposure and quantity of plastic.Mechanisms of vascular absorption and transport.Cardiac bioaccumulation and pathogenesis.Dose and time-dependency of cardiotoxicity.

In conclusion, cardiovascular diseases are the major cause of death worldwide, and exploring the impact of plastic particles on mortality and morbidity could potentially save lives. A synergic role for ecologists, epidemiologists, clinicians, and researchers is needed to relate cardiovascular risk to increased exposure to environmental contaminants, including MPs/NPs.

## Figures and Tables

**Figure 1 biomedicines-11-00264-f001:**
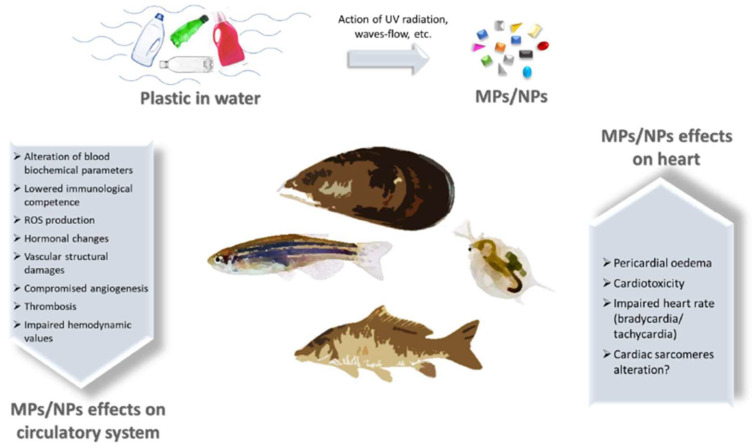
Scheme of MPs and NPs effects on cardiovascular system of aquatic organisms. Plastic wastes in water are prone to segregation due to the concerted action of UV-radiation, waves and water flows, leading to the generation of MPs and NPs. These particles reach aquatic fauna and enter the body, where they exert a plethora of detrimental effects on circulatory system and heart.

**Figure 2 biomedicines-11-00264-f002:**
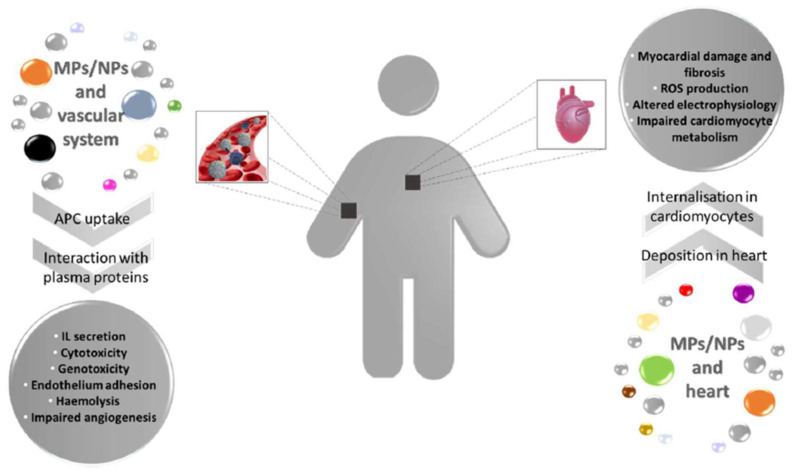
MPs and NPs effects on vascular system and heart in mammals. Plastic fragments enter the human body through different routes and are internalised within the cells of vascular system where they can trigger several cellular responses. Additionally, MPs/NPs reach the cardiac tissue causing structural and metabolic damages. IL = Interleukin.

**Figure 3 biomedicines-11-00264-f003:**
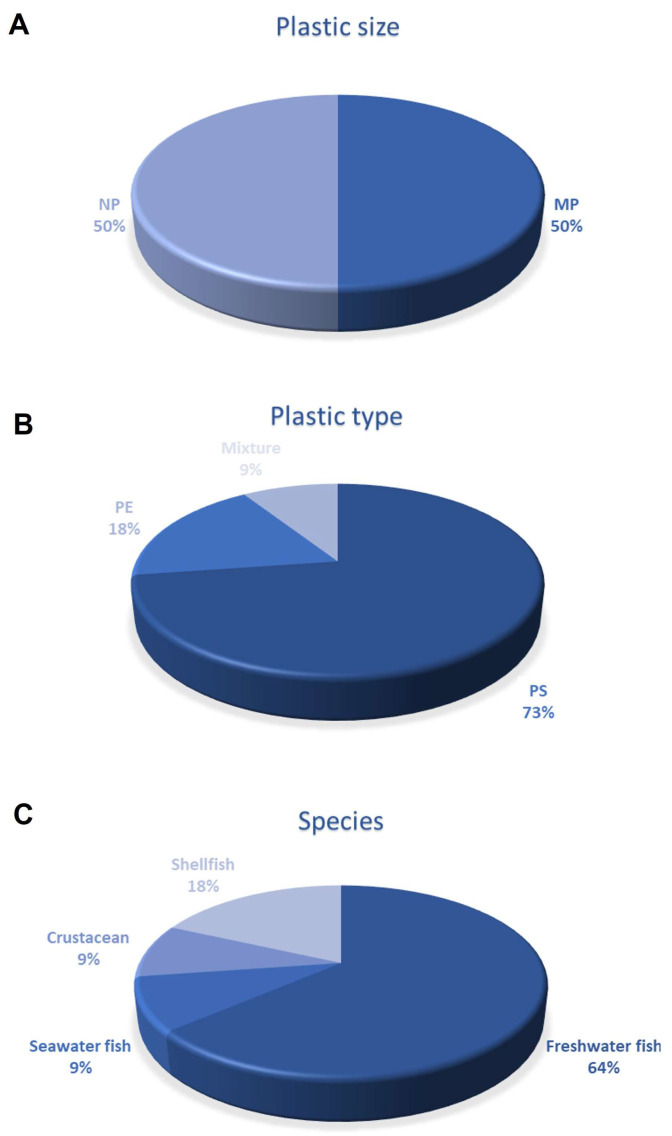
Charts summarising the studies on aquatic fauna included in the present review. Panel (**A**) reports plastic size, either MPs or NPs used in the studies. In (**B**), plastic types. In (**C**), aquatic species considered for the investigation of MPs-NPs effects.

**Figure 4 biomedicines-11-00264-f004:**
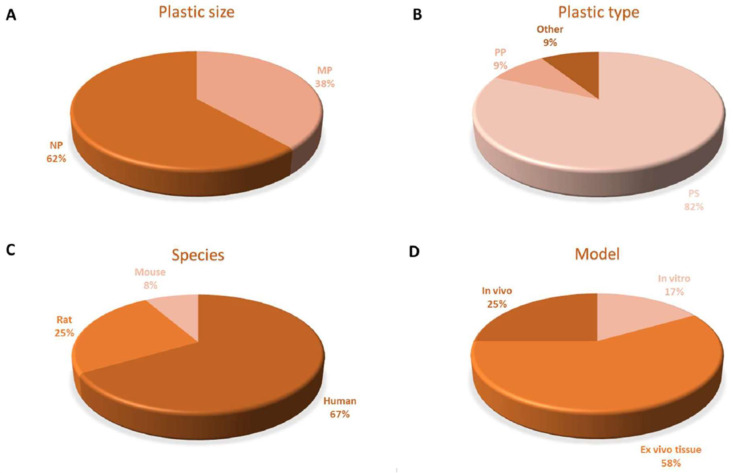
Charts describing the state-of-the-art of MPs/NPs on cardiovascular system in mammals. (**A**) Plastic particles size used in the studies here reported. (**B**) Types of plastics considered. (**C**) Species described for MP-NP effects and (**D**) experimental models used for investigation.

**Table 1 biomedicines-11-00264-t001:** Summary of the main features of MPs/NPs in studies on the cardiovascular (CV) system of aquatic organisms included in the present review. PS: polystyrene; PE: polyethylene; AChE: acetylcholinesterase; GGT: gamma glutamyl-transferase; AST: aspartate aminotransferase; ALT: alanine aminotransferase; ALP: alkaline phosphatase; LDH: lactate dehydrogenase; ACH50: alternative complement activity.

Organism	Type of Particles	Size of Particles	Observed Effects on CV System	Reference
Zebrafish (*Danio rerio*)	PS	5 μm and 70 nm	Bioaccumulation in gills (transfer to capillaries)	[44]
Red tilapia (*Oreochromis niloticus*)	PS	0.1 μm	Bioaccumulation in gills and transfer to capillaries	[45]
Blue mussel (*Mytilus edulis)*	PS	3.0 or 9.6 μm	Transfer to capillaries, internalisation into haemocytes	[46]
Common carp (*Cyprinus carpio*)	Mixture (mainly PE)	Mixed	Reduced plasma levels of AChE and GGT, and increased AST, ALT, ALP and LDH, lowered lysozyme and ACH50 activities, lowered total immunoglobulins and complement C3 and C4 factors	[48]
Marine medaka (Oryzias melastigma)	PS	10 μm	Bioaccumulation in gills, ROS production and histopathological changes in loco	[49]
Zebrafish (*Danio rerio*)	PE	Mixed (191.10 ± 3.13 nm)	Vascular endothelium damage and compromised angiogenesis, pro-thrombotic state. Altered hemodynamic	[50]
Mediterranean mussel (Mytilus galloprovincialis)	PS	50 nm	Blood cells apoptosis, compromised immunocompetence	[51]

**Table 2 biomedicines-11-00264-t002:** Main characteristics of MPs/NPs and effects on cardiac tissue of aquatic fauna. PE: polyethylene; PS: polystyrene.

Organism	Type of Particles	Size of Particles	Observed Effects on Heart	Reference
Zebrafish (*Danio rerio*)	PE	Mixed (191.10 ± 3.13 nm)	Pericardial oedema	[50]
Zebrafish (*Danio rerio*)	PS	42 nm	Bradycardia	[53]
Zebrafish (*Danio rerio*)	PS	51 nm	Bioaccumulation in pericardium, bradycardia	[54]
Marine medaka (Oryzias melastigma)	PS	10 μm	Bradycardia	[49]
Goldfish (*Carassius auratus*)	PS	70 nm and 5 μm	Tachycardia	[55]

**Table 3 biomedicines-11-00264-t003:** Summary of MPs/NPs characteristics and effects on CV system of mammalian. PBMC: Peripheral Blood Mononuclear Cells; RBCs: red blood cells; HUVEC: Human Umbilical Vein Endothelial Cells; PS: polystyrene; PP: polypropylene; PMN: Polymorphonuclear leukocytes.

Specimen/Model	Type of Particles	Size of Particles	Observed Effects on Circulating Cells and Vasculature	Reference
Human serum	PS	80–170 nm	Internalisation into monocytes, granulocytes and myeloid dendritic cells	[56]
Human whole blood	PS	0.04–0.09 μm	Internalisation into white cells and monocytes, genotoxic effects on PMN and monocytes	[57]
Human PBMCs, murine macrophages	PS	20, 100, 200, 500 and 1000 nm	Internalisation into macrophages and phagocytes (for coronated plastics), IL-6 release	[58]
Human plasma	PS	100 nm	Lymphocytes and erythrocytes promoting cytotoxicity and genotoxicity, haemolysis. Escape immune surveillance (corona formation)	[60]
Human RBCs	PS	49.9 ± 6.3; 107.9 ± 1.4; 243 ± 3.0 nm	Aggregation and adhesion to endothelial cells (more pronounced with decreasing size of NPs)	[61]
Sheep RBCs, human PBMCs, murine macrophage cell line, human mast cell line, human fibroblasts	PP	20 or 25–200 μm	Haemolysis, pro-inflammatory cytokines (IL-2, IL-6, TNF-α) release	[63]
Sprague-Dawley rats	PS	24 μm	Vascular occlusion, hypercoagulability, pulmonary embolism	[65]
HUVEC	PS	0.5, 1, and 5 μm	Impaired angiogenesis (through inhibition of VEGF pathway), autophagy and necrosis	[66]

**Table 4 biomedicines-11-00264-t004:** Table summarising the main features and effects of MPs/NPs on mammalian cardiac tissue and cardiomyocytes. PS: polystyrene; LTCC: L-type calcium channel.

Specimen/Model	Type of Particles	Size of Particles	Observed Effects on Heart	Reference
Sprague-Dawley rats	PS	24 μm	Arterial hypotension, increased arterial lactate concentration, hypoxia	[65]
Specific pathogen free (SPF) CD-1^®^ mice	PS	0.7918 ± 0.00273 and 0.7939 ± 0.00282 μm	Particles accumulation	[69]
Fischer 344 rats	PS	50 nm	Particles accumulation	[70]
Sprague-Dawley rats	PS	20 nm	Particles accumulation (for both maternal and foetal heart)	[73]
Wistar rats	PS	0.5 μm	Particles accumulation, internalisation into cardiomyocytes. Myocardium apoptosis, fibrosis. Increased myocardial creatine-kinase MB and cardiac Troponin I, oxidation	[75]
Sprague-Dawley rats	PS	50 nm	Particles internalisation into cardiomyocytes. Inhibition of LTCC and decreased contraction forces	[76]

## Data Availability

Not applicable.

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
