# Peer review of "Microplastics: A Matter of the Heart (and Vascular System)"

_biomedicines, 2023, doi:10.3390/biomedicines11020264_

Round 1

Reviewer 1 Report

The paper submitted by Persiani et al. reviews the latest studies concerning the effects of plastic particles on the health of different types of aquatic organisms and finally on humans. 

The manuscript is well organized, clear and the authors have proposed some pertinent perspectives. However, some minor modifications are necessary:

1. the authors might add some additional information about the type of plastic found in aqueous environment. which are the first three types of polymers found?

2. the authors have described the toxic effects of plastic, especially PS, on different organs but it will be of interest to know at which concentration these effects were observed. in fact, it will be important to have some information about different parameters used in these studies. 

Reviewer 2 Report

This literature review is very well written and is easy to read and understand.

The theme of this review is very current and relevant, with the number of articles on this subject being very small, greatly increasing the value of this review. The images chosen and taken by the authors are also very pertinent and help to summarize the most relevant mechanisms on the effect of this pollutant on the cardiovascular system.

Thus, I am of the opinion that the article should be accepted, however I think that it could be improved with the presence of summary tables, namely the MPs/NPs effects on cardiac cells and in the vascular cells.

Reviewer 3 Report

The action of microparticles on the organisms of vertebrates and invertebrates is presently in the focus of interest as more than a decade ago the effects of nanoparticles. The submitted review article aims to describe the effect of microplastic on the cardiovascular system. To provide additional knowledge on this topic, it is essential that the submitted text highlights the difference between the action of microplastic and nanoplastic and differentiates between the sizes.

General comments

Despite the analytical problems to estimate the number of nano- and microparticles in the environment, information on the relation of micro- and nanoplastics in the environment is essential to assess the risk of microplastic particles for the health of animals and humans.

A review on microplastic should address common aspects of micro- and nanoparticles and highlight size-dependent differences. Along with the reported findings, the sizes of the particles, which were studied in the respective publications, should be stated in text and/or table.

There should more information/data to which extent the microplastic particles can reach the systemic circulation when exposed by oral, inhalation or dermal route. The differences in the extent of uptake between the exposed animals is important to understand inter-species differences and the relevance of animal data for human exposure.

As the authors suggest by the title, the action on the heart cannot be regarded isolated from effects on blood and blood vessels. It is therefore important that effects of the particles on thrombosis (coagulation, platelets, endothelial activation) are included.

It is correct that the adsorption of pollutant to the particles may play a role in the biological action of particles. It would be good to include data/estimations on the capacity of the microplastic particles to transport pollutants and to include examples for relevant pollutants.

What is the importance of the composition of the microplastic particles (polystyrene, polypropylene, etc.).

No information on the potential contribution of consumption of plants is given.

Round 2

Reviewer 3 Report

The authors addressed my comments.